# Bi-Axial Woven Tiles: Interlocking Space-Filling Shapes Based on Symmetries of Bi-Axial Weaving Patterns

Vinayak R. Krishnamurthy*
J. Mike Walker '66 Department
of Mechanical Engineering
Texas A&M University

Ergun Akleman†
Departments of Visualization &
Computer Science and Eng.,
Texas A&M University

Sai Ganesh Subramanian‡
J. Mike Walker '66 Department
of Mechanical Engineering
Texas A&M University

Katherine Boyd§
Department of Visualization
Texas A&M University

Chia-An Fu¶
Department of Visualization
Texas A&M University

Matthew Ebert‖
J. Mike Walker '66 Department
of Mechanical Engineering
Texas A&M University

Courtney Starrett**
Department of Visualization
Texas A&M University

Neeraj Yadav††
Department of Architecture
Texas A&M University

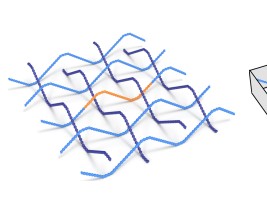 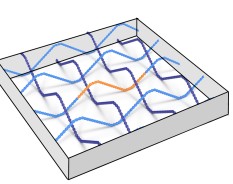 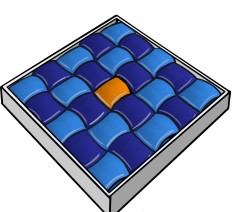 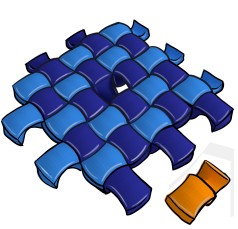 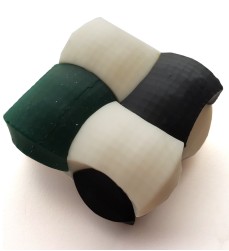

(a) *A set of curve segments that are closed under symmetry operations. The yellow curve shows the basic element of the repeating curve segment for this case.*

(b) *The curve segments in a 2.5D fundamental domain, which is rectangular prism.*

(c) *Voronoi decomposition of fundamental domain using curve segments as Voronoi sites. Yellow tile in the center is a space filling tile.*

(d) *Assembly of space filling tiles by its replicas. The yellow tile is removed to show the inner structure.*

(e) *Physical assembly of 3D printed tiles in a different configuration, where flexible dark green piece plays the role of locking this configuration.*

Figure 1: *The computational pipeline for the geometric design and fabrication of woven tiles is shown. This particular example illustrates the tiles generated using the plain weave symmetries filling 2.5D space. The Figure 1c shows the curves in fundamental domain. The yellow curve shows the basic element of the repeating curve segment. All other curve segments in the fundamental domain can be obtained by rotating and translating this yellow curve. The Figure 1d shows overall assembly by removing the tile that corresponds to yellow curve. We obtained the shapes of top surfaces also with Voronoi decomposition.*

## Abstract

In this paper, we introduce a geometric design and fabrication framework for a family of interlocking space-filling shapes which we call bi-axial woven tiles. Our framework is based on a unique combination of (1) Voronoi partitioning of space using curve segments as the Voronoi sites and (2) the design of these curve segments based on weave patterns closed under symmetry operations. The underlying weave geometry provides an interlocking property to the tiles and the closure property under symmetry operations ensure single tile can fill space. In order to demonstrate this general framework, we focus on specific symmetry operations induced by bi-axial weaving patterns. We specifically showcase the design and fabrication of woven tiles by using the most common 2-fold fabrics called 2-way genus-1 fabrics, namely, plain, twill, and satin weaves.

---

*e-mail: vinayak@tamu.edu
†e-mail: ergun.akleman@gmail.com
‡e-mail: sai3097ganesh@tamu.edu
§e-mail: katherineboyd@tamu.edu
¶e-mail: sqree@tamu.edu
‖e-mail: matt_ebert@tamu.edu
**e-mail: cstarrett@tamu.edu
††e-mail: nrj31y@tamu.edu

**Index Terms:** Human-centered computing—Visualization—Visualization techniques—Treemaps; Human-centered computing—Visualization—Visualization design and evaluation methods

## 1 INTRODUCTION

### 1.1 Motivation

Space-filling shapes have applications in a wide range of areas from chemistry and biology to engineering and architecture [48]. Using space-filling shapes, we can compose and decompose complicated shell and volume structures for design and architectural applications. Space-filling shapes that are also tileable, can be further provide an economical way for constructing structures because they can be mass-produced. Despite their practical importance, the variety of 2.5D and 3D space-filling tiles at our disposal are quite limited. The most commonly known and used space-filling shapes are usually regular prisms such as rectangular bricks since they are relatively easy to manufacture and are widely available. However, reliance on regular prisms, significantly constrains our design space for obtaining reliable and robust structures [16, 45, 58, 70, 71], particularly when current additive manufacturing techniques are gradually becoming more affordable across engineering and construction domains. In this paper, we introduce a geometric design and fabrication framework for a new class of interlocking space-filling shapes which we call bi-axial woven tiles.

Systematic design of modular, tileable and, simultaneously interlocking building blocks is a challenging task. We find that there is

currently no principled approach that would allow one to generate such building blocks. To this end, we present a general conceptual framework that takes as input a set of curves determined through fabric weave patterns and uses these curves as Voronoi sites to partition space. This allows one to decompose space into any arbitrary partition induced by the input curves wherein each partition can be considered as a tile.

## 1.2 Inspiration & Rationale

While our framework is general, we specifically chose bi-axial weave patterns to demonstrate our approach. The inspiration for using bi-axial weave patterns came from the fact that woven fabrics can form strong structures from relatively weak threads through interlacing (or interlocking) [49]. Therefore, woven structures have been known to have several applications ranging from textiles to composite materials. Using this as our rationale, we intended to investigate the possibility of constructing tiled assemblies of interlocking space-filling shapes that leverage the thread interlacing process from woven fabrics, specifically 2-fold structures. The advantage of choosing weave patterns is that they are closed under symmetry operations thereby allowing us to systematically and intuitively design and construct an entire family of interlocking space-filling shapes — bi-axial woven tiles. In addition to providing simple and intuitive control, woven tiles also relate to the structural characteristics of woven fabrics, which have been known to have several applications ranging from textiles to composite materials.

In addition to the potential advantages rooted in mechanical behavior, 2-fold fabric structures are particularly useful for our purpose because of their geometric simplicity and intuitiveness. They provide a simple approach for designing interlocking space-filling tiles. A particular subset of 2-fold fabrics, known as 2-way and genus-1, are particularly useful for simple and intuitive control. They can be constructed using regular square grid as the guide shape and they include most popular weaving structures such as plain, twill, and satin.

## 1.3 Summary of Approach

Using the properties of 2-fold 2-way genus-1 fabrics, our approach is to obtain desired curves segments that are closed under symmetry operations. One simplification of these fabrics is that each curve segment can be chosen to be planar (see Figure 1a). In addition, we can define all well-known fabric patterns such as plain, twill and satin using only three parameters. The fundamental domain of these symmetry operations is a prism with a square base because of 2-way and genus-1 property (see Figure 1b). In other words, we only have to compute Voronoi decomposition of the fundamental domain. Then, the Voronoi region of the curve segment in the center, shown as yellow in Figure 1a, is used as the space-filling tile.

We present a simplified method to compute Voronoi decomposition of fundamental domain with these curve segments. We first sample each curve segment to obtain a piece-wise linear approximation. We compute 3D Voronoi decomposition for each sample point. This process gives us a set of convex Voronoi polyhedra for the same curve segment. The union of these convex polyhedra gives us desired space filling tile. We identify simple and robust algorithms to take union of all convex Voronoi polyhedra that comes from the same piece-wise linear curve segment. We also developed a tile beautification process inspired by the fact that the points of equal distance to a planar surface and a line parallel to the surface lie on a parabolic cylinder. We add two planar surfaces that sandwich the control curves from top and bottom also as Voronoi sites. Resulting Voronoi decomposition automatically provides nice boundaries that consist of parabolic regions. The 2D equivalent of the idea is shown in Figure 2.

To demonstrate our approach, we have designed many interlocking and space-filling tiles. We call them woven tiles since 2-fold

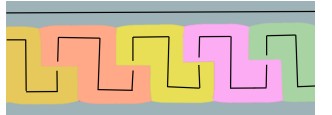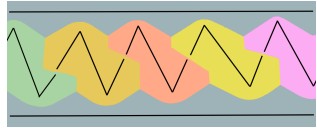

(a) *Voronoi decomposition two enclosing lines with S shape pieces that forms a square wave.*

(b) *Voronoi decomposition two enclosing lines with S shape pieces that forms a triangular wave.*

Figure 2: *A 2D example of beautification of boundaries. Note that inclusion of two enclosing lines allows to create curved outer boundaries in Voronoi decomposition. The effect is more visible with interaction of the sharp corners of triangular wave. In 3D, since we use a surface and curve, we obtain curved boundaries as ornament.*

fabrics refers woven structures. This terminology is also helpful since we can use weaving terminology to describe the variety of tiles produced by this approach as plain, twill or satin woven tiles. Because of their symmetry properties, these tiles can be assembled in more than a single configuration. Some assembly structures can even create loops as shown in Figure 1e. For these cases, we have shown that it is possible to lock the pieces using one flexible piece.

## 1.4 Our Contributions

Our overarching contribution in this work is a general conceptual framework for generating space-filling and interlocking tiles based on the fundamental principles of fabric weave patterns in conjunction with space decomposition using 3D Voronoi partition. Based on this framework, we make four specific contributions as listed below:

1. We use our general framework to develop a simple and intuitive methodology for the design and construct *Bi-Axial Woven Tiles*, space-filling tiles derived from the symmetries induced by woven fabrics The basic idea is to use curves representing 2-way 2-fold weaving patterns (such as plain, twill, and satin) as Voronoi sites for decomposing 3-space.

2. We introduce a simple and effective algorithm for approximating the Voronoi decomposition of space with labelled curve segments as the Voronoi sites. The algorithm uses a simple process that first discretizes a curve segment into a sequence of points and then constructs a Voronoi cell of the curve simply by computing the union of *constitutive* Voronoi cells for each point on the curve. The first advantage of this method is its simplicity — it allows us to directly use standard Voronoi cell computation for points for curves. Secondly, it allows for an elegant computation of the Voronoi cell surface as a triangle mesh using a simple topological operation — removing the internal polygonal faces of adjacent constitutive cells of points.

3. We demonstrate several cases of *Bi-Axial Woven Tiles* and demonstrate techniques for the fabrication and assembly these tiles. We show the fabrication these tiles with a variety of materials (plastic, wax, and metal) by using different 3D printing, molding, and casting techniques. Furthermore, we demonstrate that these tiles can be assembled more than single configuration. From the same group, it is even possible to obtain two assemblies with different chirality (i.e. mirrored versions of each other).

4. Finally, we present a comparative structural evaluation of plain, twill, and satin tile assemblies. The finite element analyses (FEA) of these assemblies under under planar and normal loading conditions reveal that weaving allows distribution of planar and normal loads across tiles through the contact surfaces, generated with our methodology. We describe the qualitative relationship between the symmetries induced by the weave patterns to the stress distribution in the tiled assemblies.

## 2 RELATED WORK

### 2.1 Space filling Polyhedra

Space filling polyhedra, which can be used to tessellate (or decompose) a space [37], are defined as a cellular structure whose replicas together can fill all of space watertight, i.e. without having any voids between them [48]. While 2D tessellations and 2D space filling tiles are well-understood [37], problems related to 2.5D and 3D tessellations and tiles (i.e. shell and volume structures respectively) are still perceived as difficult. The perception of difficulty of 3D tessellations probably comes from the belief that tetrahedron can fill space since 500 BC. In fact, many failed efforts were made to prove this widespread belief [57].

It is now known that the cube is the only space filling Platonic solid [25]. This partly explains the widespread use of regular prisms as space filling tiles. We are indebted to Goldberg, whose exhaustive cataloguing from 1972 to 1982, helped us to access all known space-filling polyhedra [26–28]. We now know that there are only eight space-filling convex polyhedra and only five of them have regular faces, namely the triangular prism, hexagonal prism, cube, truncated octahedron [72, 73], and Johnson solid gyrobifastigium [7, 43]. Five of these eight space filling shapes are "primary" parallelohedra [14], namely cube, hexagonal prism, rhombic dodecahedron, elongated dodecahedron, and truncated octahedron. Space filling polyhedra is still active research area in mathematics [56]. However, as far as we know there exist no space filling shapes that can also interlock.

### 2.2 Interlocking Structures

History is rich with examples of puzzle-like interlocking structures, which is analyzed under the names such as stereotomy [20–22], nexorades [9, 10, 17] and topological interlocking [11, 18, 19, 68]. One of the most remarkable examples of interlocking structures is the Abeille flat vault, which is designed by French architect and engineer Joseph Abeille [23, 62]. Nexorades, which are also called the Leonardo grids, are types of structures that are constructed using notched rods that fit into the notches of adjacent rods [54, 59].

### 2.3 Geometry and Topology of Fabric Weaves

We observe that many interlocked structures can be viewed as knots and links that are decomposed into curve segments. This view simplifies the design process since we can build our framework by borrowing concepts directly mathematics literature. It can also provide significant intuition for the design since bi-axial textile weaving structures, which are also called 2-fold 2-way fabrics, also form knots and links by viewing them as structures embedded on a toroidal surface [29, 30]. The word 2-way, which is usually called *biaxial*, means that the strands run in two directions at right angles to each other- warp or vertical and weft or horizontal. The word 2-fold means there are never more than two strands crossing each other [6].

The popularity of 2-fold 2-way fabrics comes from the fact that the textile weaving structures are usually manufactured using loom devices by interlacing of two sets of strands, called warp and weft, at right angles to each other (see Figure 3a). Since the warp and weft strands are at right angles to each other, they form rows and columns. We colored warp thread blue and weft threads yellow to differentiate these two threads as shown in Figure 3c). The names warp and weft are not arbitrary in practice. In the loom device, the weft (also called filling) strands are considered the ones that go under and over warp strands to create a fabric. The basic purpose of any loom device is to hold the warp strands under tension such that weft strands can be interwoven. Using this basic framework, it is possible to manufacture a wide variety of weaving structures by raising and lowering different warp strands (or in other words by playing with ups and downs in each row).

There was no formal mathematical foundation behind bi-axial weaving until Grunbaum and Shephard, who are known by their

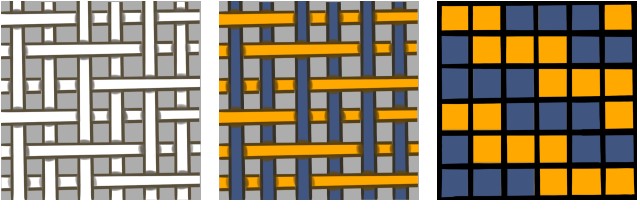

(a) *Top view of a weave with uncolored threads.* (b) *Same weave with colored warp and weft threads.* (c) *Matrix view of the same weave.*

Figure 3: The fundamental domain of 2-way 2-fold fabrics is a rectangle and they can be represented as a simple matrix. The warp threads are colored blue and weft threads are colored yellow to differentiate the two threads in the final matrix.

contributions to 2D tiling [36, 38], investigated the mathematical properties of bi-axial weaving in 1980's in a series of papers [33–35, 37]. By viewing weaves as matrices as shown in Figure 3c, they simplified the problem of classifying and analyzing woven structures.

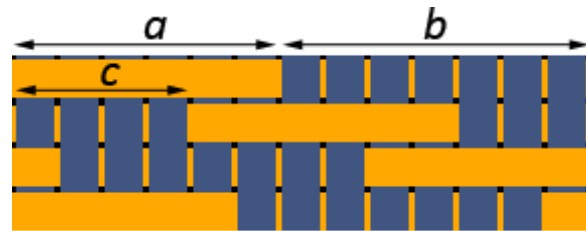

Figure 4: Three parameters, $a, b$ and, $c$, are sufficient to define all of the important 2-fold, 2-way genus-1 fabrics

Grunbaum and Shephard studied a subset of 2-fold 2-way patterns that have a transitive symmetry group on the strands of the fabric, which they called isonemal fabrics [34]. They identified all isonemal patterns that *hang together* for periods up to 17 [33, 34]. Roth [55], Thomas [64–66] and Zelinka [75, 76] and Griswold [31, 32] theoretically and practically investigated symmetry and other properties of isonemal fabrics. The identification of the *hanging-together* property is simpler for a certain type of isonemal fabrics that are called genus-1 [34]. Genus-1 means that each row with length $n$ is obtained from the row above it by a shift of $c$ units to the right, for some fixed value of the parameter $c$. Genus-1 fabrics includes two special and well known isonemal fabrics, twills and satins. A twill pattern is the one each row of a design is obtained from the row above it by a shift of one square in a fixed direction (either left or right). A satin pattern is the one in which each row or column has only one blue square in the fundamental domain given by $n \times n$ matrix. To further simplify the design, we assume that the row with length $n$ consists of $a$ number of consecutive weft (yellow) threads and $b = n - b$ number of consecutive warp (blue) threads as shown in Figure 4.

These $[a, b, c]$-fabrics are guaranteed to hang together if $n = a + b$ and $c$'s are relatively prime [35]. The most widely used fabric pattern, plain weaving, is given as $[1, 1, 1]$-fabric using $[a, b, c]$ notation. Twills are given as either $[a, b, 1]$ or $[a, b, -1]$. Satins are described by $b = 1$ and $c^2 = 1 mod(a + b)$ [12]. The genus-1 isonemal fabrics described by the $[a, b, c]$ notation not only include well-known patterns such as plain, twill, and satin but also a wide variety of additional bi-axial weaving patterns as shown in Figure 5. For instance, for the $[3, 3, 2]$ pattern shown in the figure, the notation $[a, b, c]$ can

represent a non-fabric that can fall-apart. Fortunately, as discussed earlier it is easier to avoid the non-fabrics that can fall-apart unlike a general isonemal weaving case. We can simply check whether $n = a + b$ and $c$'s are relatively prime or if any row or column has no alternating crossing [35]. In conclusion, the $[a,b,c]$ notation provides a simple process to design control curves for bi-axial woven tiles. Figure 5 also demonstrates that among the $[a,b,c]$ patterns, the pattern is rotation-invariant only for plain, twill, and satin cases. This is because in plain, twill, and satin cases, warp and weft patterns are guaranteed to be mirrored versions of each other [37]. Since this is required to obtain a single tile, we focus on only plain, twill and satin woven tiles in this paper.

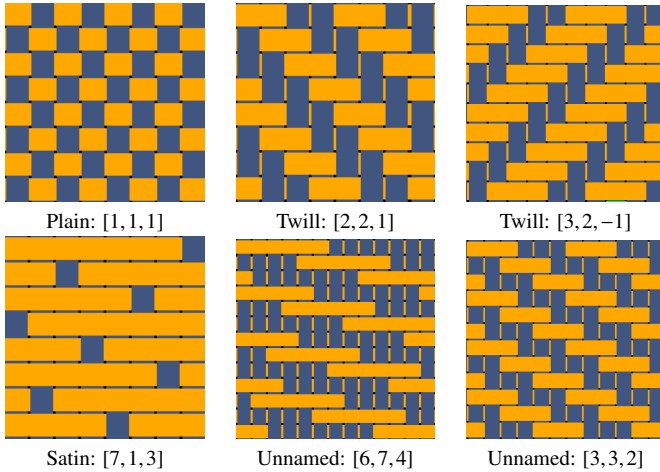

| Plain: [1,1,1] | Twill: [2,2,1] | Twill: [3,2,−1] |
| Satin: [7,1,3] | Unnamed: [6,7,4] | Unnamed: [3,3,2] |

Figure 5: Examples of isonemal genus-1 patterns that can be represented by three parameters shown in Figure 4. Unnamed pattern [6,7,4] hang together, but [3,3,2] falls apart since $3 + 3$ and $2$ are not relatively prime.

## 3 THEORETICAL FRAMEWORK

To our knowledge, none of the existing approaches for producing interlocking structures currently provide space-filling pieces simultaneously. For instance, Leonardo grids are simply finite cylinder shapes that leave most space empty. Our approach in this paper is to fill (or decompose) the space appropriately using Voronoi decomposition. It appears that the concept of filling space using Voronoi decomposition actually came from Delaunay's original intention for the use of Delaunay diagrams. He was the first to use symmetry operations on points and Voronoi diagrams to produce space filling polyhedra, which he called Stereohedra [15, 56]. Recent work on Delaunay Lofts extended points to specific types of curves to obtain more complicated space filling structures [60]. In this paper, we first observe that any shape (a line, a curve, or even a surface) can be used as a Voronoi site to fill the space. If our initial configurations of the shapes are "good" such as being closed under symmetry operations, we are guaranteed to obtain interesting decomposition of the space — this is the real premise of this paper.

The essential conceptual contribution of allowing any type of shapes as Voronoi sites is the extension of potential space filling shapes from simple polyhedra to almost any shape with curved edges and curved faces. In fact, allowing curved edges and faces significantly extends the design space of space-filling polyhedra. For instance, Escher's complicated 2D space filling tiles have been created by using curved edges [41, 51]. Another recently developed space filling shapes, called Delaunay Lofts [60] extended the design space by allowing curved edges and curved faces. Allowing any

type of shapes as Voronoi sites not only enables a systematic search of desired shapes from large number of potential candidates, but also provides **a simple design methodology** to construct space filling structures.

Based on this point of view, the key parameters for the classification of space-filling shapes are essentially the topological and geometric properties of Voronoi sites and their overall arrangements that are usually be obtained by symmetry transformations (rotation, translation, and mirror operations). The types of shapes and transformations uniquely determine the properties of the space decomposition. Now, based on this view point, let us again look at Stereohedra and Delaunay lofts.

For Stereohedra, the shapes of Voronoi sites are points, 3D $L_2$ norm is used for distance computation, underlying space is 3D and any symmetry operation in 3D are allowed [15, 56]. Based on these properties, we conclude that Stereohedra can theoretically represent every convex space filling polyhedra in 3D. Since the points are used as Voronoi sites and $L_2$ norm is used, the faces must be planar and edges must be straight in the resulting Voronoi decomposition of the 3D space.

For Delaunay lofts, on the other hand, the shapes of Voronoi sites are curves that are given in the form of $x = f(z)$ and $y = g(z)$, for every planar layer $z = c$ where $c$ is a real constant, a 2D $L_2$ norm is used to compute distance, underlying space is 2.5 or 3D and only 17 wallpaper symmetries are allowed in every layer $z = c$ [60]. Based on these properties, we conclude that Delaunay lofts (1) consists of a stacked layers of planar convex polygons with straight edges, and (2) in each layer there can be only one convex polygon. In Delaunay lofts the number of sides of the stacked convex polygons can change from one layer to another. In conclusion, the faces of the Delaunay lofts are ruled surfaces since they consist of sweeping lines. Edges of the faces can be curved.

For bi-axial woven tiles in this paper, the shapes of Voronoi sites are curve segments obtained by decomposing planar periodic curves that are given -essentially[1]- in the form of $z = F(x+n) = F(x)$ and $z = G(y+n) = G(y)$, where $n = a + b$ the period of fabric, where $F$ can be any periodic function as far as it consists of $a$-length up regions and $b$-length down regions as shown in Figure 6. The function $G$ is just the mirror of $F$ with $a$-length down regions and $b$-length up regions. The curve segments are obtained from these two periodic functions by just restricting its domain into a region such as $(x_0, x_0 + kn)$. These curve segments are closed under symmetries of bi-axial weaving patterns, that are given by $90^0$ rotation and translation operations. 3D $L_2$ norm is used for distance computation. Underlying space is normally 2.5D, i.e. a planar shell structure [1].

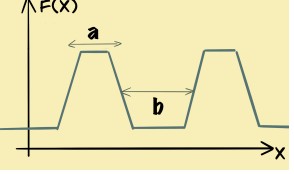 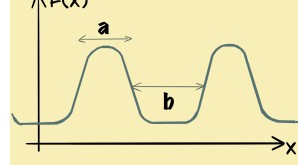

(a) *A piece-wise linear periodic curve.*   (b) *A derivative continuous periodic curve.*

Figure 6: Examples of periodic curves that can be used as Voronoi sites, i.e. control curves.

Based on these properties, it is clear that the resulting tiles would usually be genus-0 surfaces with curved faces and edges. Because of its bi-axial property, the fundamental domain for these tiles would always be a rectangular prism, an extruded version of the original rectangular fundamental domain of corresponding 2-way 2-fold

---
[1]We actually use parametric curves. This is only for providing a quick and simple explanation without a loss of generality

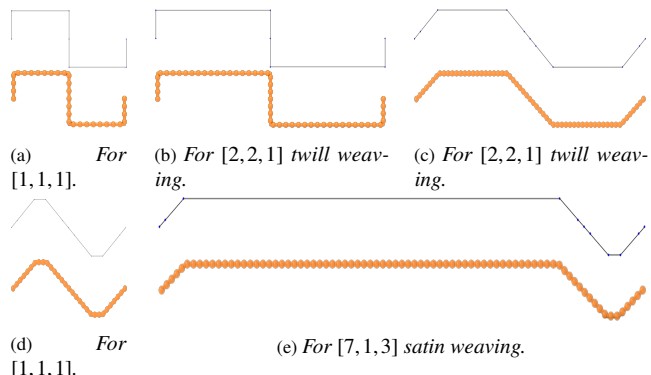

(a) *For* [1,1,1].  (b) *For* [2,2,1] *twill weaving.*  (c) *For* [2,2,1] *twill weaving.*

(d) *For* [1,1,1].  (e) *For* [7,1,3] *satin weaving.*

Figure 7: The basic degree-1 NURBS curves that are used to construct woven tiles. Each curve is created by changing positions of 11 control points. The figures at the top are actual curves. The figures at the bottom are points that are created by sampling the initial curves. These points that approximate the curves are used as Voronoi sites.

fabric [39]. Therefore, the tiles that perfectly decompose this rectangular prism domain will also fill all 3D space.

## 4 DESIGN AND FABRICATION PROCESS

Our bi-axial woven tile design process consists of three steps: (1) Designing curve segments; (2) Designing 3D configuration of the curves segments to be used as Voronoi sites; and (3) Decomposition of the space using Voronoi tessellation. For all steps, we have used the simplest approaches which simplify the design process and provides robust computation.

### 4.1 Designing Curve Segments

We designed our control curves by using Non-Uniform Rational B-Splines (NURBS). We initially allowed the higher degree curves to allow $C^1$ and $C^2$ continuity, but, quickly realized that piecewise-linear curves are sufficient to obtain desired results for woven tiles. Therefore, we designed all curves with degree 1 NURBS. For all cases, we use the same 11 control points. We simply move the positions of the control points to obtain the curve segments for desired weaving pattern as shown in 7. To construct these curves, in addition to three weaving parameters, i.e. $a$, $b$, and $c$, we provide one additional control: the angle of connection of two consecutive tiles. By changing the angle we can obtain **Square Waves**, which appears to be binary function such as the ones shown in Figures 7a and 7b, and **Partly Triangular Waves**, which appears to be regular piece-wise linear such as the ones shown Figures 7d, 7c and 7e. The two consecutive tiles produced by square waves can sit at the top of each other as shown in Figures 11a and 11b. With partly triangular tiles, we can adjust this angle as shown in Figures 11d and 11c and 11e.

### 4.2 Designing Voronoi Sites

Based on three weaving parameters, i.e. $a$, $b$, and $c$, we have developed an interface to create 3D curve segments that are closed under symmetry operations of 2-fold 2-way genus-1 fabrics. The algorithm consists of three stages be given as follows:

1. Create initial curve segment as $x = F_x(t)$, $y = 0$ and $z = F_z(t)$ based on $a$ and $b$ values, and curve type. Without loss of generalization, assume $t \in [0,1]$, $z \in [0,1]$, and $x \in [-n/2, n/2]$. Note that $n = a + b = F_x(1) - F_x(0)$.

2. Create two replicas of the curve and translate them along the $x$ axis by adding and subtracting its period $n = a + b$ respectively. This creates three copies of initial curve that follows each other.

3. Create two replicas of of these three curves. Translate one of them using $(c, 1, 0)$ vector and translate the other $(-c, -1, 0)$. This translation operation must be done in modulo $3n$.

   - **Remark 1:** This operation creates a $3n \times 2 \times 1$ rectangular prism domain, which is sufficient to compute tiles. Note that we assume the height of the curves is 1 unit.

   - **Remark 2:** This rectangular domain is not a fundamental domain of the curve symmetries. It is only applicable for genus-1 case.

4. Create perpendicular curve segments.

5. **Remark 3:** Perpendicular curve segments are guaranteed to be the same for plain, twill and satin. Therefore, we only focus on thise to obtain single tile.

In practice we create these curves in a larger rectangular domain as shown in Figures 8, 9, and 10 to see the structure of the curves better. These rectangular domains must be larger than the $3n \times 2 \times 1$ domain we described earlier to guarantee we obtain at least one tile that can fill the space. In other words, at least one curve must be covered with its neighboring curves to guarantee that the Voronoi region that corresponds that particular curve segment fill the space. In Figures 8, 9, and 10, which shows two plain, two twill and one satin cases, the center curve is colored yellow. We have implemented this interface by using SideFX's Houdini, which is a robust 3D software that provides a node-based system for fast and easy interface development.

### 4.3 Decomposition of the Space

Accurate decomposition of a given space using curves as Voronoi sites can be quite complicated. We, therefore, have developed a simple method that provides us reasonably good approximation of decomposition. Our method consist of four stages:

1. Sample the original curve segments by obtaining the same number of points for each curve segment.

2. For beautification step, create and sample two sandwiching (or bounding) planes. If not, skip this step. All the examples in this section are created using beautification step.

3. Label points as follows:

   - The points that are originated from central yellow curve are labeled using one label, say 0.

   - All other points are labeled using another label, say, 1.

   - **Remark 1:** If the beautification step is used, the points coming from the sandwiching planes are also labeled 1.

4. Decompose the space using 3D Voronoi of these points, which gives us a set of labeled Voronoi regions, which are convex polyhedra that are labeled either 0 or 1.

5. Take union of all Voronoi regions labeled 0 to obtain desired space filling tile. Union operation consists of only face removal operations as follows:

   - Remove the shared faces of two consecutive convex polyhedra coming from two consecutive sample points on the curve.

   - **Remark 2:** These faces will always have the same vertex positions with opposing order.

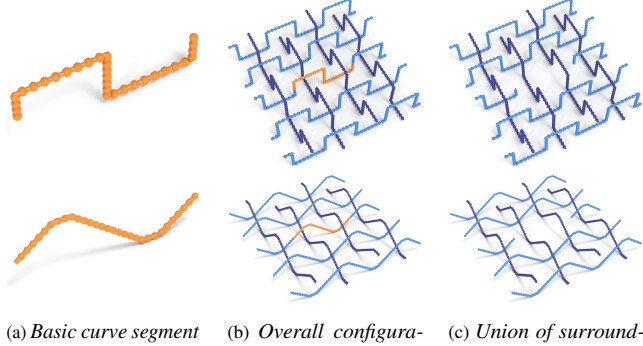

(a) *Basic curve segment in 3D for [1,1,1] plain weaving.*

(b) *Overall configuration for decomposition of rectangular prism domain.*

(c) *Union of surrounding curves provides mold structure.*

Figure 8: An example for designing control curves for [1,1,1] plain woven tiles.

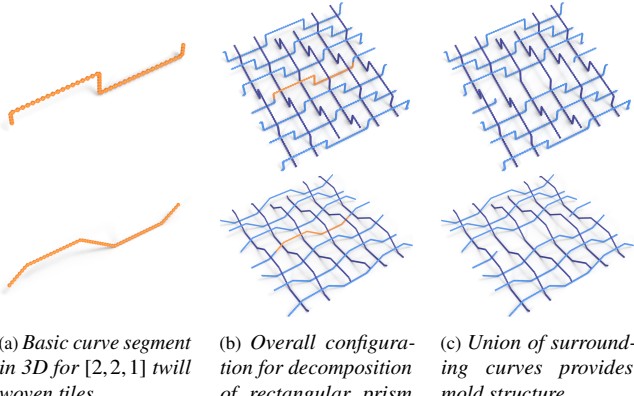

(a) *Basic curve segment in 3D for [2,2,1] twill woven tiles.*

(b) *Overall configuration for decomposition of rectangular prism domain.*

(c) *Union of surrounding curves provides mold structure.*

Figure 9: An example for designing control curves for [2,2,1] twill woven tiles.

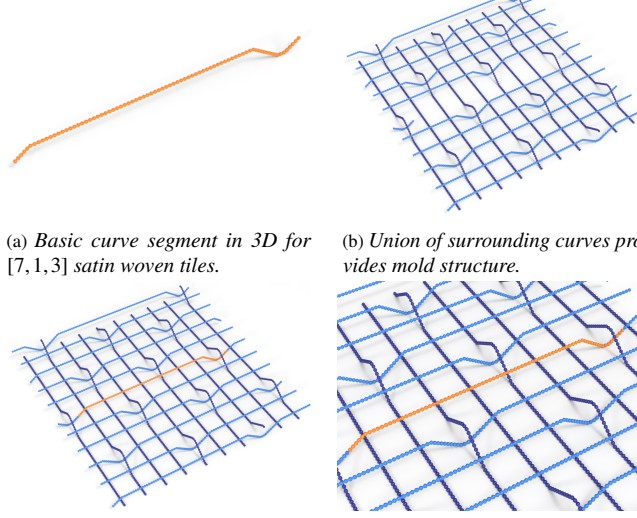

(a) *Basic curve segment in 3D for [7,1,3] satin woven tiles.*

(b) *Union of surrounding curves provides mold structure.*

(c) *Overall configuration for decomposition of rectangular prism domain.*

(d) *Close up of overall configuration.*

Figure 10: An example for designing control curves for [7,1,3] satin woven tiles.

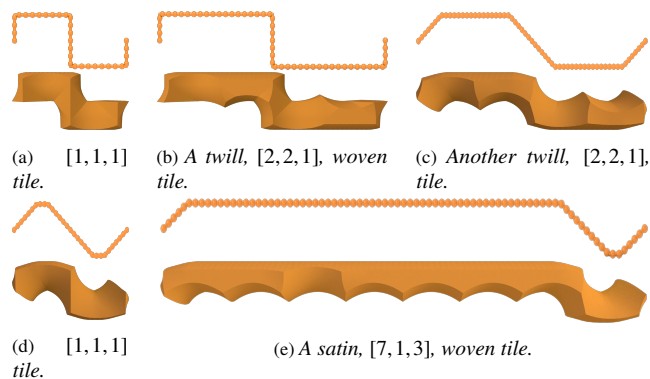

(a)  [1,1,1] *woven tile.*

(b) *A twill, [2,2,1], woven tile.*

(c) *Another twill, [2,2,1], tile.*

(d)  [1,1,1] *tile.*

(e) *A satin, [7,1,3], woven tile.*

Figure 11: Examples of plain, twill and satin woven tiles using the basic degree-1 NURBS curves shown in Figure 7.

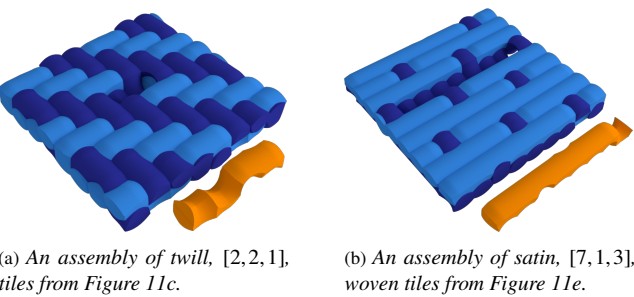

(a) *An assembly of twill, [2,2,1], tiles from Figure 11c.*

(b) *An assembly of satin, [7,1,3], woven tiles from Figure 11e.*

Figure 12: Examples of assemblies that show only the tiles cut to stay in rectangular domain.

- **Remark 3:** If underlying mesh data structure provides consistent information, this operation is guaranteed to provide a 2-manifold mesh. Even if the underlying data structure does not provide consistent information, the operation creates a disconnected set of polygons that can still be 3D printed using an STL file.

- **Remark 4:** If the beautification step is skipped, i.e. two sandwiching planes are not used, take an intersection with bounding rectangular prism.

6. **Optional Step:** Take union of Voronoi regions with label 1 to obtain a hollow space that correspond to the mold that can be used to mass produce space filling tiles. Note that we need to take again an intersection with bounding rectangular prism if the beautification step is skipped.

We have implemented this stage in both in Matlab and Houdini. For 3D Voronoi decomposition of points, we used build in functions available in Matlab and Houdini.

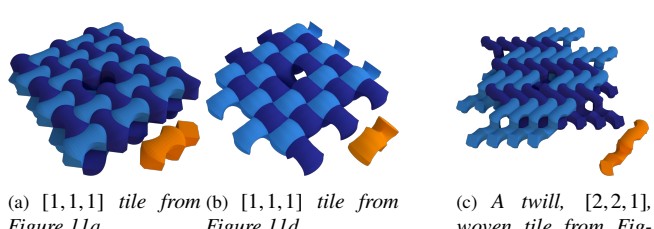

(a) [1,1,1] *tile from Figure 11a.*

(b) [1,1,1] *tile from Figure 11d.*

(c) *A twill, [2,2,1], woven tile from Figure 11b..*

Figure 13: Examples of assemblies with uncut tiles .

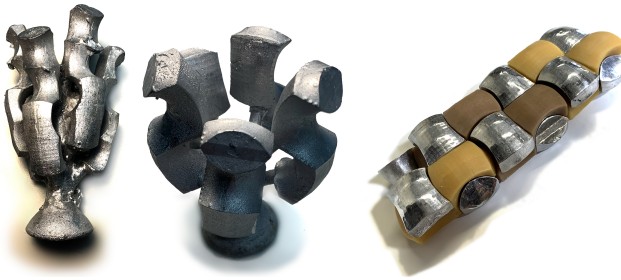

(a) *Casting aluminum tiles as a group.* (b) *Wax and aluminum tiles assembled together.*

Figure 14: Examples of casting aluminum using lost wax method.

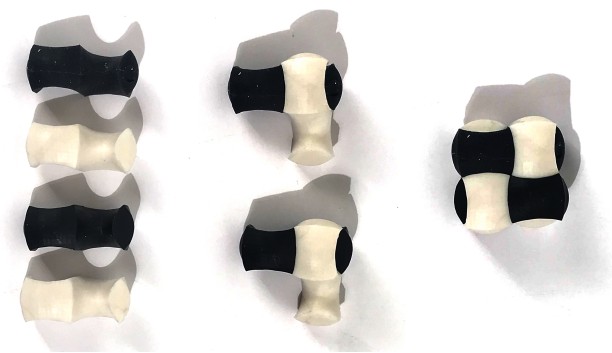

(a) *Individual plain tiles.* (b) *Plain tile pairs.* (c) *Complete assembly.*

Figure 15: Assembly of plain woven tiles. One of the black pieces is a flexible silicone piece and is needed to successfully assemble plain tiles.

## 4.4 Fabrication

All examples in this section are created using beautification step. We have printed the tiles shown in Figures 11d, 11c and 11e using both standard resin and elastic resin. For the purpose of investigation of various material properties and potential manufacturing options we made rubber molds of the tiles shown Figures 11d, and 11c for casting silicon rubber and wax versions. The wax tiles were used to cast aluminum tiles via the lost wax casting process as shown in Figure 14a. Also shown in Figure 14b is the assembly of wax and aluminum tiles.

## 5 PHYSICAL ASSEMBLY

The geometry and topology of weaves has a rich research history with several open questions relating to the ability of the weaves to *hold together*. The works by Grunbaum et al. [37] assume that the threads being woven are infinitely long. This, obviously is not the case with woven tiles, making it more difficult to completely and formally characterize the assembly of woven tiled. Therefore, our first evaluative step was to physically assemble common weaving patterns (plain, twill, and satin), with the goal to explore how the symmetries induced by these patterns affect the method of creating assemblies of the respective tiles. We are particularly interested in two aspects of woven tile assembly: (a) locking ability which maps to the *holding-together* property of the weaves and (b) chiral configurations of woven tile assemblies.

## 5.1 Locking Ability of Woven Tiles

The topology of a weaving pattern directly affects the locking ability of its corresponding woven tile. For instance, plain weave tiling results in self-locking configurations (Figure 15) identical to a plain

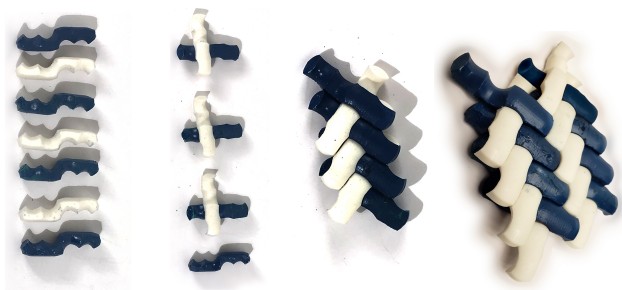

(a) *Individual twill tiles.* (b) *Twill pairs.* (c) *Twill assembly.* (d) *Twill assembly with one repetition.*

Figure 16: Assembly of twill woven tiles.

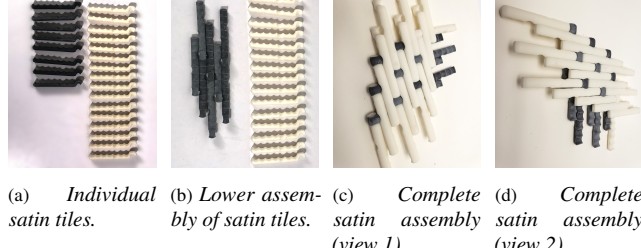

(a) *Individual satin tiles.* (b) *Lower assembly of satin tiles.* (c) *Complete satin assembly (view 1).* (d) *Complete satin assembly (view 2).*

Figure 17: Assembly of satin woven tiles.

woven fabric. Therefore, if zero tolerance is assumed, plain woven tiles cannot theoretically be assembled together with tiles constructed out of rigid materials such as PLA or Aluminium. In the $2 \times 2$ plain woven tile assembly shown in Figure 15a , one of the two black tiles (also the dark green tile in Figure 1e) is a compliant tile made of silicone, constructed through casting. This assembly is structurally stable and the geometry of the elements itself holds the structure together. Specifically, both the assembly and disassembly of the plain woven tiling is possible only through the application of force. In addition to introducing a flexible element, we also experimented with all four pieces cast in wax as well as Aluminium. In this case, the shrinkage in the individual pieces allowed for the tiling to be assembled (Figure 14b).

In case of twill weaves, we do not encounter the locking problem. As seen in Figure 16, the twill assembly can be simply created by an alternating placement of tiles along each of the axis (the white and blue tiles represent each axis). Therefore, neither the assembly nor disassembly require any application of force and we did not need any flexible pieces for twill (Figure 16c). There are two observations we make here. First, in the plain woven tiling, exactly half of each tile is above one adjacent tile and the other half is underneath a second adjacent tile. Second, in case of twill assembly, the unit tiles do share this alternate *above-underneath relationship* with their neighbors. However, note that if two twill woven tiles are combined to create a double-length tile (Figure 16d), we obtain the *above-underneath* relationship that will likely produce a perfectly interlocking tiling (thereby needing flexible tiles akin to the plain-woven case).

In case of satin weaves (Figure 17), we come to similar conclusions — there is a minimal number of repetitions of each tile to ensure a tightly packed interlocked assembly. While we can say for certain that the number of repetitions must be higher than twill, we currently do not claim what the number of repetitions should be. We believe that much work needs to be done in order to develop a formal theory for locking ability of woven tiles.

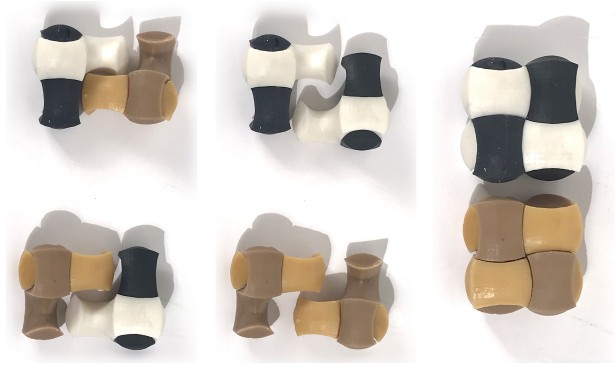

(a) *Chiral pairs of plain woven tiles cannot be assembled.*  (b) *Pairs belonging to the same chiral group can be assembled.*  (c) *Plain woven assembly of two chiral groups.*

Figure 18: An example of chirality in plain woven tile assemblies.

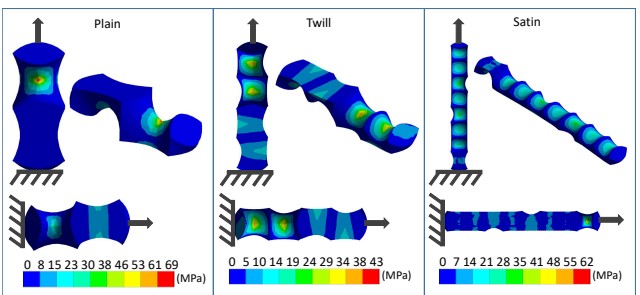

Figure 19: Von-Mises stress distribution on single woven tiles

## 5.2 Chirality

A chiral object is one that is non-superposable on its mirror image. Chirality is a fundamental to several natural phenomena and engineering applications. Our first example that explores chirality is the plain-woven assembly wherein we observed that assembling the same plain-woven tiles in mirrored configurations leads to chital assemblies (Figure 18). Penne's work on planar layouts [50] provides a formal explanation to this propery by connecting projective geometry and topology.

## 6 STRUCTURAL EVALUATION

Multiple uses of our methodology can be noted by using tiles as structural building blocks. For example, we can assemble tiles of plain weave to use them as a reinforced slab blocks. This is because, the nature of contacts between the tiles allow the forces/stresses to be distributed from one tile to other very easily. To explore this aspect better, we performed simulations of the individual shape separately and the response of the shape in the assembly. For our evaluation, we considered three commonly known plain, twill, and satin weaves and analyzed their response to basic mechanical loading conditions. The main motivation is to observe key relationships between the symmetries induced by these well-known weave patterns and the corresponding mechanical behavior.

## 6.1 Evaluation Methodology

FEA is considered as one of the most powerful tools for studying the mechanical properties of textile composites owing to the fact that the interactions between the unit cells are complex in nature [61] in addition to experimental methods [44]. Based on this, present FEA analysis with two objectives. First, we are interested in understanding the effect of contact between the interlocked woven

| | Plain | Twill | Satin |
|---|---|---|---|
| Minimum Stress (Pa) | 2099 | 4.76e-9 | 1212 |
| Average Stress (MPa) | .93 | .59 | .97 |
| Maximum Stress (MPa) | 12.11 | 48.41 | 19.64 |
| Minimum Displacement (m) | 0.00 | 0.00 | 0.00 |
| Average Displacement (m) | 2.91e-6 | 1.97e-5 | 8.36e-6 |
| Maximum Displacement (m) | 1.22e-5 | 1.97e-6 | 4.13e-5 |

Table 1: Minimum, maximum and average stresses and displacements for woven tile assemblies under normal loading.

tiles. Second, we wanted to observe the stress distributions for an individual woven tile and patterns that emerge as an effect of the weaving pattern.

We used the ANSYS Workbench 2019 R1 and conducted static structural analysis for all simulations. We specifically explored two loading conditions: planar (tensile load applied on the plane of the tiled assembly) and normal (compressive load applied normal to the plane of the tiled assembly). For simplicity, we assume unit forces (1 N) and moments (1 N-m) across all simulations. All the dimensions were chosen in accordance with the 3D printed shapes. For the analysis, we first imported a given woven tile as a solid body in SolidWorks 2019 and created assemblies of these tiles. Here, the size of the assembly was an important factor for a fair comparison. We used the satin weaves as the benchmark since it required the largest number of tiles. Based on this, we created an $8 \times 8$ assembly for plain, twill, and satin woven tile assemblies. For materials considerations, we used the material properties of PLA (Polylactic acid). Specifically, we set the density to $1250 kg/m^3$, Young's modulus to $3.45 * 10^9$ Pa and the Poisson's Ratio to 0.39. We further assume all the contact regions to be friction-less. For each loading condition, Von-Mises stress and the total deformations were evaluated.

## 6.2 Tensile Loading on Single Tiles

We begin with a simple test of tensile loading of individual unit tiles for plain, twill, and satin cases (Figure 19). The first key observation for the same 1N load is that the twill woven tile admits the minimum value of the maximum stress (43 MPa) as compared to the plain and satin tiles (69 MPa and 62 MPa respectively). What is commonly evident across all cases is that the maximum stresses are located at the saddle points of the tiles. Also note that these are the critical points where any two tiles come in contact for a given assembly causing a high stress region. Therefore, the individual stress contours (Figure 19) help us identify the critical high stress regions through which load is internally transferred from part to part in a woven assembly. Furthermore, since the critical stresses are transferred through two doubly-curved surfaces in contact, the stress can be propagated in multiple directions.

## 6.3 Common Behavioral Characteristics Across Weaves

The deformation of the weave assembly behaves similar to a solid block of material for both planar and normal loading conditions. This is expected since the tiles are space filling and interlocking. The stress characteristics of the shapes, however, was found to have a deeper relationship with the the geometry of the weave pattern (Figure 21 and 20). Below, we make observations regarding these distributions for normal and planar loading conditions.

## 6.4 Assembly Under Normal Loading

Threads of plain, twill and satin were created by joining individual tiles. These threads were assembled to form an $8 \times 8$ assembly (Figure 20). The faces on the perimeter were fixed and a unit normal force was applied perpendicular to the plane of the assembly in the central region.

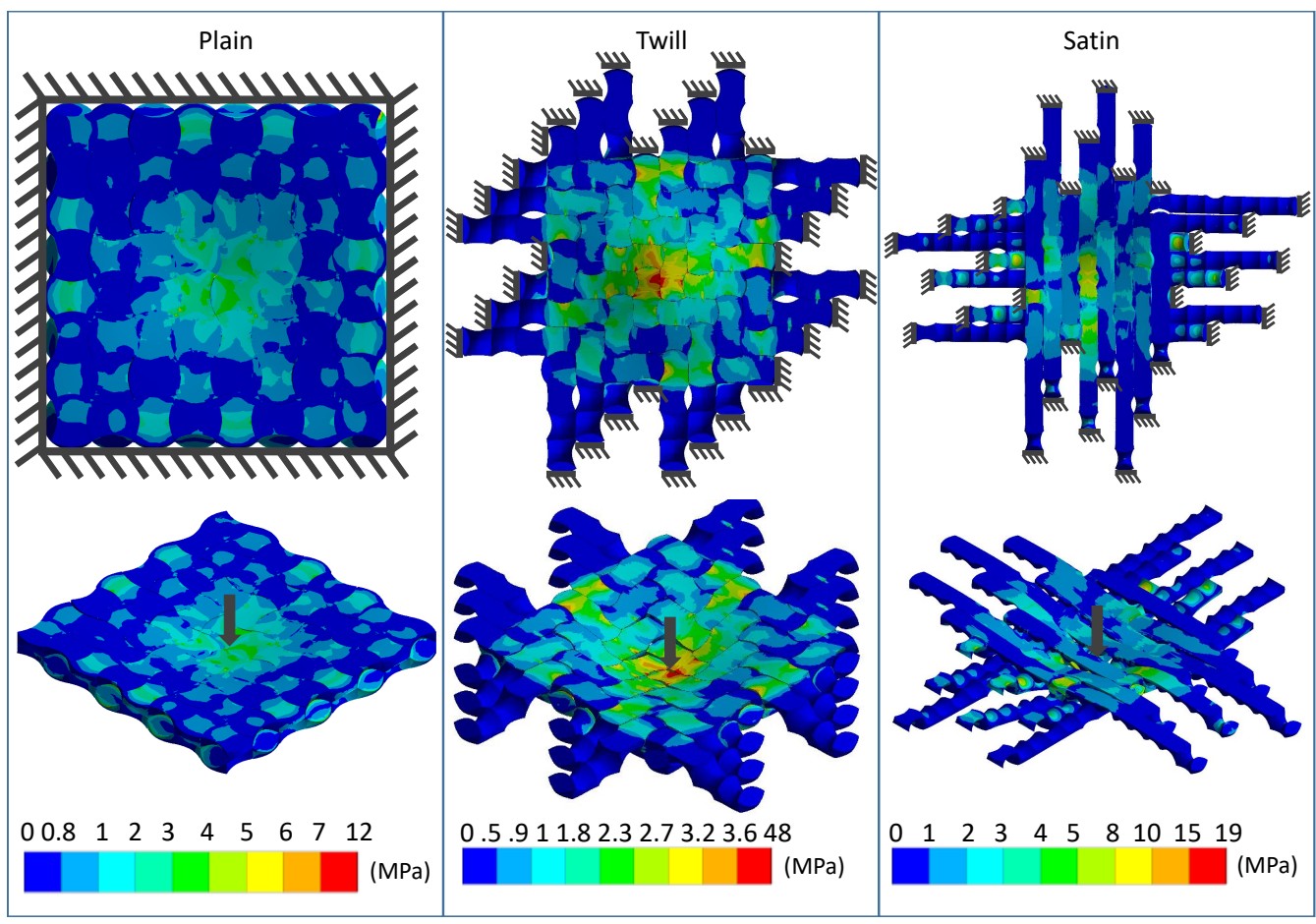

Figure 20: Von-Mises stress distributions and deformation of 8x8 assemblies of plain (left), twill (middle) and satin symmetries under normal loading. The assemblies are based on threads constructed out of multiple tiles. A uniform unit normal force was applied in the central region of the assembly indicated by the direction of arrow. The faces in the region indicated by the ground symbol were assigned as fixed supports.

We observe that the woven nature of the tiles allows the deformation caused by the central load to be distributed uniformly around the axis of application in the assembly. Given that the deformation of the assembly is similar to a solid block, the contacts between the treads along the two axes transfer the incoming forces and stresses to the adjacent tiles. The distribution for each assembly respects the geometry of the weave pattern. For example, in the plain woven case (Figure 20 left panel), the stress starts as maximum at the center tiles and assumes a radial checkerboard pattern as it shifts toward the periphery. For instance, for plain woven assembly, we observe a transition from a concentrated stress distribution to a radial checkerboard pattern reminiscent of the original plain woven symmetries (Figure 5 top-left image). Interestingly, in twill we notice that stress propagates in a spiral-like manner (Figure 20 middle panel) from the central region.

From a cursory numerical comparison (Table 1), there are two interesting observations to be made. First, the twill assembly has the lowest average stress (0.59 MPa) while the plain and satin assemblies exhibit comparable average stresses . This is supported in textile engineering literature [63]. However, what is interesting that twill simultaneously exhibits the largest range of stress $(0-48$ MPa) when compared to plain $(0-12$ MPa) and satin cases. This implies that while twill may be better suitable for load-sensitive applications, plain and satin assemblies may be better candidates for scenarios that need durability.

### 6.5 Assembly Under Planar Loading

In case of planar loading, we applied tensile load on threads in one axial direction and observed the effect the orthogonal threads. We notice that the average and maximum stress values were lower in the assembly compared to individual simulations for all weaving patterns. This means that the assembly allows for an efficient stress distribution.

Similar to stress patterns in normal loading, the planar simulations show a pronounced stress arrangement. While the plane of maximum stress distribution is orthogonal to the load direction in plain woven tiles, we note that in case of twill it is aligned approximately perpendicular to the weave direction as also noted by previous works [8, 42]. The plan of maximum stress is in fact orthogonal to the diagonal lines produced on the face if the assembly is developed into a twill fabric. In the contrary, no apparent stress patterns can be noticed in satin assembly. We also note a significantly high value of maximum stress in satin compared to twill and plain (Table 2). This also agrees with the fact that in clothing satin, being a long yarn floats makes it unstable [63] and needs a much densely woven fabric to counter this.

The twill assemblies again exhibit the lowest average Von Mises stress levels (0.42 MPa) as compared to plain (0.57 MPa) and satin (3 MPa). This is aligned with current literature on weave mechanics wherein plain and twill weaves display superior resistance to tensile loading in comparison to satin weaves because of lower frequency of alternation between the two axes [63]. Similar to nomal loading, twill also exhibits maximum range of stress values $(0-23$ MPa) when compared to plain $(0-10$ MPa) and satin $(0-793$ MPa) tile assemblies. Here, once again, satin clearly demonstrates significantly high maximum stress when compared to plain and twill as is evident from the topology of the satin weave.

### 7 DISCUSSION

The work presented in this paper provides (1) many new directions that need to be explored further; and (2) many interesting questions that need to investigated further. In the rest of this section, we discuss some of the future directions to explore and some of the questions to investigate.

|  | Plain | Twill | Satin |
|---|---|---|---|
| Minimum Stress (Pa) | 308 | 2.92e-7 | 2.31e-5 |
| Average Stress (MPa) | .57 | .4217 | 3.08 |
| Maximum Stress (MPa) | 10.43 | 23.32 | 793.80 |
| Minimum Displacement (m) | 0.00 | 0.00 | 0.00 |
| Average Displacement (m) | 1.54e-6 | 9.27e-6 | 2.27e-6 |
| Maximum Displacement (m) | 4.14e-6 | 7.91e-6 | 1.50e-5 |

Table 2: Minimum, maximum and average stresses and displacements for woven tile assemblies under planar loading

### 7.1 Generalization based on Knots and Links

2-fold fabric structures are much richer than just 2-way genus-1 fabrics. Their real power can be best understood with extended graph rotation systems (EGRS) that was introduced in early 2010's [5, 6]. EGRS allows us to use orientable 2-manifold meshes as guide shapes to represent knots and links. The guide shapes help us to classify the fabrics. For instance, the guide shapes for 2-fold 2-way fabrics are regular grids embedded on genus-1 surfaces. For 2-fold 3-way fabrics, we need regular hexagonal or regular triangular grid embedded on genus-1 surfaces [6]. This is useful since some of the Leonardo grid designs are based on also 3-way woven patterns [54]. Using regular maps [13, 67], it is also possible to obtain hyperbolic tiling. Using the regular maps that correspond to hyperbolic tiles as guide shapes, 2-fold k-way genus-n fabrics can be obtained. From these fabrics, one can also obtain space filling shapes. For practical applications, there is a need for a significant amount theoretical work.

### 7.2 Locking

The key open question that we hope to answer in our future work is a formally supported computational methodology for determining minimum tile repetition to generate pure interlocking of woven tiles. Here, Dawson's work on the enumeration of weave families can provide an important starting point as a means to develop such a method based on sound mathematical principles. We see that the locking ability of woven tiles is related to three interlinked concepts in geometry and topology literature, namely, liftability [53], oriented matroids [77], and planar layouts of lines in 3-space [50]. To simply determine repetitions for locking is only the first step. Once we obtain a locking configuration, the second challenge is to determine the minimum number of flexible/compliant elements to make the assembly possible. We only showed this example for the plain woven tiles (Figure 1e). To the best of our knowledge, a general strategy for this problem is currently unavailable.

### 7.3 Chirality

As we have seen in our results, chirality is a key aspect of further investigation in this research. A recent work discovered how to produce handedness in auxetic unit cells that shear as they expand by changing the symmetries and alignments of repeating unit cells [46]. Using the symmetry and alignment rules we can potentially expand our woven tiles to develop a new class of rigid and compliant structures [40, 52, 74]. Recent works on knot periodicity in reticular chemistry [47] and tri-axial weaves (also known as mad weaves) [24] are fundamental examples of how the geometry and physics of chirality are connected. Thus, identifying any fundamental multi-physical behavior of the assemblies shown in this work and beyond would allow us to construct assemblies with several practical applications such as mechanically augmented structures in mechanical, architectural, aerospace [2, 3], and materials [69] engineering. The main gap that must first be filled, however, is a complete characterization of chirality of woven tiles including and beyond plain, twill, and satin varieties.

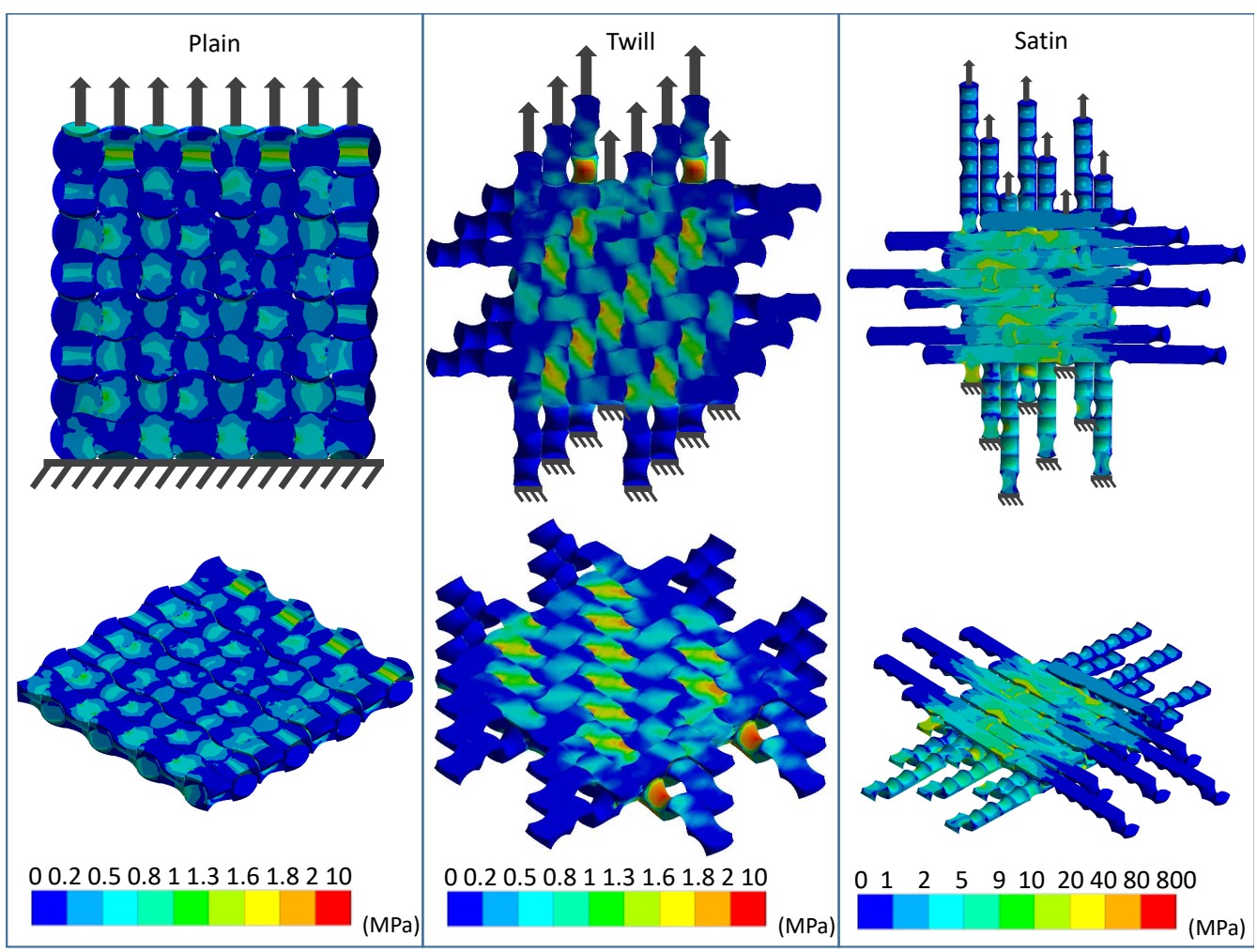

Figure 21: Von-Mises stress distributions and deformation of 8x8 assemblies of plain (left), twill (middle) and satin symmetries under planar loading. The assemblies are based on threads constructed out of multiple tiles.

## 7.4 Structural Behavior & Topology

We observed correlation between the weave topology and the structural behavior of woven tile assemblies. However, our analysis and observations are currently qualitative. Therefore, a formal and constitutive methodology for connecting the topology and structural properties is an important future direction that needs attention. As an important example, determining the relationship between direction of stress distributions to the weave parameters (the numbers $a$, $b$, and $c$ in Figure 4) will allow for systematic design of woven tiles for specific applications.

## 8 Conclusion & Future Directions

In this paper, we have developed a methodology to design interlocking space-filling tiles that we call *bi-axial woven tiles* that are generated using the topology of bi-axial woven fabrics. To this end, we developed a method to create desired input curves segments using the properties of 2-fold 2-way genus-1 fabrics. We further developed a simple method to compute Voronoi decomposition of the curve segments. We demonstrated our general methodology by designing, fabricating, assembling, and mechanically analyzing woven tile assemblies. We 3D-printed some of these tiles and physically observed their mathematical and physical properties. We also developed molds to directly cast these shapes with a wider range of materials such as silicone and aluminium. While our physical evaluation of the individual and assembled properties of these tiles aligns with the current literature on woven fabrics, we show some interesting additional properties that were not previously apparent. Furthermore, our results suggest that interlocking these tiles have potential to replace existing extrusion based building blocks (such as bricks) which do not provide interlocking capability.

We want to point out that 2-fold fabrics are not really a final frontier. It is also possible to represent k-fold fabrics using 3-manifold meshes as guide shapes [4]. The extension to k-fold fabrics requires even more theoretical foundations, but it demonstrates the potential. A significant advantage of using guide shapes is that the topological properties of the knotted structures do not change with any geometric perturbation of the guide shapes. In conclusion, even though we chose our proof of concept tiles from 2-fold 2-way types, the ideas in this paper can be extended into more general types of fabrics with the maturation of theoretical work in regular maps and 3-manifold meshes.

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
