# OpenReview forum: "Bi-Axial Woven Tiles: Interlocking Space-Filling Shapes Based on Symmetries of Bi-Axial Weaving Patterns"
_graphicsinterface.org/Graphics_Interface/2020/Conference — GI 2020_

### Official Review · AnonReviewer2 · 2020-04-20
**Nicely written paper on an interesting problem.**

**Rating:** 9
**Confidence:** 3

**Review:**

The paper presents a technique to design shapes that can tile space seamlessly. The authors build on the theory of weaving patterns and use them to define a space partitioning via Voronoi diagrams with 1-dimensional sources.

While the theory of wallpaper groups in 2d  is well known, I was not aware of the analogous 3d problem. The paper does an excellent job at introducing the problem and also describes the theory of weaving patterns in a very compelling way. The practical algorithm is well motivated and seems to be very solid. I also like the FEA analysis part which gives an idea of  the physical properties of the designs.

It would have been nice to see some more practical examples where such patterns could be employed. From a practical perspective I would have liked to see a clearer focus on patterns that form an interlocking assembly without the need of compliant elements.

I believe this paper should be accepted due to its novel content, thorough evaluation and convincing presentation. This paper could also inspire further research leading to practical applications in architecture, design and fabrication in general.

---

### Official Review · AnonReviewer1 · 2020-04-21
**Very thought provoking paper!**

**Rating:** 9
**Confidence:** 3

**Review:**

This paper proposed an approach to design tiles that interlock and fill space in two and a half dimensions (height field). It is claimed that prior work (primarily theoretical) focused on interlocking shapes, or space filling shapes, but not both. The space filling property is achieved by building on Delaunay's Stereohedra and Delaunay Lofts. The interlocking property is achieved by building on the theory of genus-0 2-way 2-fold (biaxial weave) fabric. Combining these two theories appears to me to be novel, and exciting. An interface allows the user to create 3D curve segments that are closed under the symmetry operations of a 2-way 2-fold genus-0 fabric. The generated designs are also actually fabricated (in various materials and with various methods), and the structural mechanical behaviour is studied for the different weave patterns (nice!). Physical locking is demonstrated in the fabricated examples. The role of chirality is briefly explored and demonstrated.

The paper builds on (and nicely attributes) Voronoi's Stereohedra, space-filling polyhedra designed by applying symmetry operations to generating points of a Voronoi diagram.

Building on the recent "Delaunay Lofts," the present paper allows for any line/curve/surface to serve as the Voronoi site, provided certain reasonableness conditions, e.g., that the initial configuration of the replicated shape is closed under symmetry operations.

P. 4 L. 69 and P.5 L. 7 –––the writing is confusing because they both seem to introduce Delaunay Lofts for the first time.

I found the demonstration of choice of chirality neat.

I found the discussion summarizing the work of Grunbaum and Shephard and genus-1 fabric theory to be very illuminating. Neat stuff! I'm glad I learned that.

Fig 2: Since the point of this figure is to show the beautification benefit of including the top and bottom lines, it would help to show these results side by side with those obtained without including the top and bottom lines. Note that I am unclear what that means... to exclude the top and bottom lines.... since then the regions would be infinite.

The first mention of "2-way 2-fold weaving patterns" merits one or more citations, and an explanation.

line 23: "under under"

For twill, you mention that there was no need for a flexible piece in the fabrication process. Do that mean that the material can still come apart? Presumably (from the theory) it cannot. So, maybe this can be explained further. For instance, is the twill easy to assemble / disassemble without any forces or flexibility because the boundary conditions at the border are not being imposed, but then when the boundary conditions are improved it gains its strength and stays together? What does this tell us about transfer of stress to the boundary I wonder?

This is a very thought provoking paper!

---

### Official Review · AnonReviewer3 · 2020-04-22
**Positively rated: New framework for woven structures with interesting applications to pre-fabricated structural building blocks**

**Rating:** 8
**Confidence:** 3

**Review:**

Originality and significance:
Woven fabrics have been heavily studied in computer graphics. This paper provides a different framework to classify such woven structures, considering base template curves, how to express the over-under pattern, and symmetry properties. I find the framework a bit oversold, but it does offer a clean way to parameterize weaving. The paper describes a pipeline to create space filling tiles by applying 3D voronoi decomposition, using the template curves as voronoi sites. This idea is effective but a minor contribution as it is a straightforward application of voronoi cells.

I found the structural applications compelling. The tiles could be practical for pre-fab construction, e.g. interlocking bricks rather than cast-in-place slabs. As mentioned below, this application domain should be described much earlier in the paper to better motivate the space filling tiles. FEM tests for different weave patterns are interesting to see how stress distributions are affected (see questions below). The paper mentions reinforced slab blocks, but as far as I can tell reinforcement was not tested in the paper.

It would be helpful to see further discussion (or future work) on assembly order of the tiles, and how to make guarantees on locking/stability behavior.

Clarity:
The exposition could use improvement.
- The introduction is heavy in terminology (e.g. 2-fold 2-way biaxial structures) making it a dense read. I appreciated the helpful background given in Sec. 2.3 "Geometry and Topology of Fabric Weaves". Please re-summarize definitions of terminology immediately in the introduction to improve clarity.
- The practical implications of the paper did not become clear to me until Sec. 6 "Structural Evaluation". I encourage discussing applications of structural building blocks in the Introduction. Also since Sec. 6 is also a major piece of the contribution.
- Pg 4, please define "relatively prime"
- Pg 4 lin 32: typo: b=n-b

The paper is quite long. Writing could be made more concise overall. Some other areas to cut:
- Fig 8-10: The difference between figures for overall configuration and union of surrounding curves is very minor. I'd suggest removing the union of surrounding curves images to shorten the paper. It's also not very illustrative for understanding the mold structure since only the curves are shown, not the 3D geometry.
- Fig 14: The aluminum casts are nicely made, but I don't see the need for including these in the paper. What is the relevance to the research contributions?
- Sec 5.1 should be significantly reduced. Locking ability is a compelling problem, but there are no guarantees or other formal conclusions given here.

Validation:
Questions regarding the structural testing:
- Are the planar and normal load testing using the same amount of material between all pattern types? I see that an 8x8 grid is used, but there may still be variations in volume. It also looks as though extra material surrounding the 8x8 grid was included in the analysis for twill and satin.
- How do the woven structures compare to a regular flat slab of the same volume? This should be included in the results.

Overall I find this a strong paper for GI with interesting contributions at the intersection of weaving and structural analysis. But exposition should be improved to properly reflect the contributions and improve readability.

---

### Meta-Review · Area_Chair1 · 2020-04-22

**Recommendation:** Accept
**Confidence:** 4

**Metareview:**

This paper is likely to inspire and spur additional work. Including it clearly strengthens GI. Please incorporate the feedback noted in the reviews.

---

### Decision · Program_Chairs · 2020-04-25

Accept